

# GeNNet: an integrated platform for unifying scientific workflows and graph databases for transcriptome data analysis

Raquel L. Costa[1,2], Luiz Gadelha[1], Marcelo Ribeiro-Alves[3] and Fábio Porto[1]

[1] DEXL Lab, National Laboratory for Scientific Computing (LNCC), Petrópolis, Rio de Janeiro, Brazil
[2] National Institute of Cancer (INCA), Rio de Janeiro, RJ, Brazil
[3] Laboratory of Clinical Research in DST- AIDS, National Institute of Infectology Evandro Chagas, Oswaldo Cruz Foundation, Rio de Janeiro, Brazil

Corresponding author
Raquel L. Costa, quelopes@lncc.br, quelopes@gmail.com

## ABSTRACT

There are many steps in analyzing transcriptome data, from the acquisition of raw data to the selection of a subset of representative genes that explain a scientific hypothesis. The data produced can be represented as networks of interactions among genes and these may additionally be integrated with other biological databases, such as Protein-Protein Interactions, transcription factors and gene annotation. However, the results of these analyses remain fragmented, imposing difficulties, either for posterior inspection of results, or for meta-analysis by the incorporation of new related data. Integrating databases and tools into scientific workflows, orchestrating their execution, and managing the resulting data and its respective metadata are challenging tasks. Additionally, a great amount of effort is equally required to run in-silico experiments to structure and compose the information as needed for analysis. Different programs may need to be applied and different files are produced during the experiment cycle. In this context, the availability of a platform supporting experiment execution is paramount. We present GeNNet, an integrated transcriptome analysis platform that unifies scientific workflows with graph databases for selecting relevant genes according to the evaluated biological systems. It includes GeNNet-Wf, a scientific workflow that pre-loads biological data, pre-processes raw microarray data and conducts a series of analyses including normalization, differential expression inference, clusterization and gene set enrichment analysis. A user-friendly web interface, GeNNet-Web, allows for setting parameters, executing, and visualizing the results of GeNNet-Wf executions. To demonstrate the features of GeNNet, we performed case studies with data retrieved from GEO, particularly using a single-factor experiment in different analysis scenarios. As a result, we obtained differentially expressed genes for which biological functions were analyzed. The results are integrated into GeNNet-DB, a database about genes, clusters, experiments and their properties and relationships. The resulting graph database is explored with queries that demonstrate the expressiveness of this data model for reasoning about gene interaction networks. GeNNet is the first platform to integrate the analytical process of transcriptome data with graph databases. It provides a comprehensive set of tools that would otherwise be challenging for non-expert users to install and use. Developers can add new functionality to components of GeNNet. The derived data allows for testing previous hypotheses about an experiment and exploring new ones through the interactive graph database environment. It enables the analysis of different data on humans, rhesus, mice and rat coming from Affymetrix

platforms. GeNNet is available as an open source platform at https://github.com/raquele/GeNNet and can be retrieved as a software container with the command docker pull quelopes/gennet.

# INTRODUCTION

The passage of cellular information through the events of transcription and translation postulates the central dogma of molecular biology presented in 1958 by Francis Crick (*Crick, 1970*). Despite the knowledge of the structure of DNA and its main biological functions, it was only in the past few decades, with the advancement of high-throughput technologies, that it became possible to quantify the transcripts produced in large-scale. Since then, substantial progress has been noted, for instance, in the identification of prognostic genes and biomarkers, and in the classification and discrimination of subtypes of tumors (*Robles & Harris, 2017*; *Guinney et al., 2015*; *Alizadeh et al., 2000*; *Golub et al., 1999*; *Zhang et al., 2012b*). Currently, microarray and RNA-Seq are the main technologies available and widely used (*Zhao et al., 2014*) in the quantification of gene expression, with advantages and disadvantages in the choice and use of each of them. For instance, on one hand, in RNA-Seq one may both identify new transcripts and observe isoforms (*Conesa et al., 2016*). On the other hand, the low cost of microarrays, in relation to RNA-Seq, still makes their use very appealing for well-known organisms.

Regardless of the technology employed, the results of transcriptome analysis can be represented as a complex interaction network. In such a network, nodes represent transcripts, genes or proteins and the connections between them can be modeled by edges having a weight assigned to them. For example, in gene co-expression networks the links may represent the correlation between the genes limited by a significant value through a cut-off value (*Zhang et al., 2012b*; *Choobdar, Ribeiro & Silva, 2015*; *Zhang & Horvath, 2005*). Strong (positives or negatives) significant correlations among a group of genes may indicate elements that participate in the activation or repression of pathways or biological functions relevant to the studied phenomenon (e.g., immunity, cell differentiation, angiogenesis, etc.). In addition, the same results can be enriched with information from external biological networks such as protein interaction networks (PPI) or even information on identification of key elements in the regulatory process such as the transcription factors (*Zhang et al., 2012a*; *Mathelier et al., 2014*). The analysis of the networks may explore topological metrics determining the connectivity between the nodes, of which the most connected can be indicated as targets in molecular modeling, development of biomarkers, etc. Through complex biological networks, we can extract topological properties such as 'small-world', 'hierarchically modular' and 'scale-free network' (*Barabasi, 2009*; *Albert, 2005*).

Managing such complex network is, however, a challenge. Current approaches employ analysis and visualization software such as Cytoscape (*Smoot et al., 2011*) and

Grephi (*Bastian, Heymann & Jacomy, 2009*). While such programs make it possible to explore complex relationships between heterogeneous information in biological systems, the results of data analyses often remain fragmented. This imposes difficulties, either for posterior inspection of results or meta-analysis by the incorporation of new related data. Furthermore, the heterogeneity of biological data adds to the problem complexity (*Maule, Emmerich & Rosenblum, 2008*), as it is difficult to find a conceptual data schema that follows a fixed and strict structure, such as in relational databases. Modifying the data schema in these cases can result in conflicts or inconsistencies in a database. In the era of expanding and interconnected information, new data models appeared, such as column-oriented, key-value, multidimensional, and graph databases. These are commonly called NoSQL (*Not only SQL*) databases and often have advantages regarding scalability (*Stonebraker, 2010*). Graph-based data models, in particular, are useful for data in which the relationship between attributes is one of the most important aspects to be taken into consideration during querying. The graph database is an intuitive way for connecting and visualizing relationships. In graph databases the nodes represent objects, and the edges represent the relationships among them. Both nodes and edges can hold properties, which add information about the objects or the relationships. In recent years, this database model has been used in many bioinformatics applications and are particularly promising for biological datasets (*Preusse, Theis & Mueller, 2016*; *Johnson et al., 2014*; *Balaur et al., 2016*; *Henkel, Wolkenhauer & Waltemath, 2015*; *Muth et al., 2015*; *Lysenko et al., 2016*). Have and Jensen (*Have, Jensen & Wren, 2013*) observed that for path and neighborhood queries, Neo4j, a graph database, can be orders of magnitude faster than PostgreSQL, a widely used relational database, while allowing for queries to be expressed more intuitively.

Besides the growing need for an adequate representation of biological data, the accumulation of molecular biology data motivated the development of pipelines, scientific workflows, and platforms for analyzing data (*Shade & Teal, 2015*; *Conesa et al., 2016*). Many researchers are using these integrative approaches for analyzing metagenomes, proteomes, transcriptomes and other 'omics' data (*Joyce & Palsson, 2006*). Regardless of the 'omics' technology, there are many steps from the acquisition of raw data to the selection of a subset of representative genes that explain the hypothesis of the scientists. Combining databases and tools into computational analyses, orchestrating their execution, and managing the resulting data and its respective metadata are challenging tasks (*Ghosh et al., 2011*). Academic journals, for instance, are demanding better reproducibility of computational research, requiring an accurate record of parameters, data, and processes also called provenance (*Carata et al., 2014*), used in these activities to support validation by peers (*Sandve et al., 2013*).

Overcoming many of these challenges can be supported by designing and executing these computational analyses as scientific workflows (*Deelman et al., 2009*), which consist of compositions of different scientific applications. Their execution is usually chained through data exchange, i.e., data produced by an application is consumed by subsequent applications. Scientific Workflow Management Systems (SWMSs) enable for managing the life cycle of scientific workflows, which is usually given by composition, execution and analysis (*Liu et al., 2015*). Many SWMSs, such as Galaxy

(*Giardine et al., 2005*), Taverna (*Oinn et al., 2004*), Tavaxy (*Abouelhoda, Issa & Ghanem, 2012*) and Swift (*Wilde et al., 2011*), natively support reusing previously specified workflows (*Goble & De Roure, 2007*) and gathering provenance (*Gadelha et al., 2012*). More recently, scripting languages such as R and Python incorporated features typically available in SWMS. RDataTracker (*Lerner & Boose, 2015*), for instance, adds provenance tracking to R scripts and noWorkflow (*Murta et al., 2015*) adds the same functionality to Python. This facilitates the specification and execution of scientific workflows in scripting languages, which is the approach we use in this work. The scientific workflow we propose (GeNNet-Wf) is implemented in R and its activities are comprised of calls to functions of various R libraries, such as limma (*Smyth, 2004*), GOstats (*Falcon & Gentleman, 2007*), affy (*Gautier et al., 2004*) and WGCNA (*Langfelder & Horvath, 2008*). Integrating scientific workflows with database systems allows for managing and persisting the data manipulated in these workflows in a structured way. This allows for scientists to perform complex data pre-processing analysis and to make the resulting data available for further investigation using queries expressed in a high-level query language. This enables expressing declaratively what data is required without saying how data should be obtained. Moreover, it abstracts away from the user low-level data management details such as accessing files where contents of a database are stored (*Garcia-Molina, Ullman & Widom, 2009*). We argue that integrated web applications, involving scientific workflows and databases, can hide the complexity of underlying scientific software by abstracting away cumbersome aspects, such as managing files and setting command-line parameters, leading to increased productivity for scientists. One critical aspect of enabling reproducible computational analyses is keeping track of the computational environment components, i.e., operating system, libraries, software packages and their respective versions (*De Paula et al., 2013*).

Currently, the vast quantity of functions performed by distinct software lead to a considerable amount of time being employed in installing and configuring them, requiring users to deal with sometimes complicated installation procedures and errors related to software dependencies and versions. Virtualization is a promising technique to tackle these problems (*Daniels, 2009*). In particular, operating system-level virtualization, as provided by 'software containers', allows for running applications and services that are instantiated on isolated environments (containers) on a hosting computer system. Containers provide all the dependencies required for these applications and services to run and can be built in a programmable way to ensure that they will be composed of the same libraries and software packages every time they are instantiated. This considerably facilitates the deployment of software systems since developers can deliver software containers for their applications directly to users or data center administrators. Docker, for instance, is an open-source platform that allows for managing containers (*Merkel, 2014*; *Boettiger, 2015*). It has a container repository called Docker Hub (https://hub.docker.com) where developers can make software containers for their applications available for download. Many traditional software and tools are available on Docker Hub and it is widely used with around five billion software containers downloaded from the repository up to August, 2016 (https://blog.docker.com/2016/08/docker-hub-hits-5-billion-pulls/). In Bioinformatics, there are already tools that are available as Docker software containers and explore features

such as reproducibility (*Hung et al., 2016*; *Belmann et al., 2015*) or applications areas such as transcriptomics (*Zichen et al., 2016*). *Di Tommaso et al. (2015)* showed that containers have a negligible impact on the performance of bioinformatics applications. Other examples of software distributed as Docker software containers are available. AlgoRun (*Hosny et al., 2016*) provides a modular software container with frequently used bioinformatics tools and algorithms that can be accessed through a browser or a Web application programming interface. ReproPhylo (*Szitenberg et al., 2015*) implements a phylogenomics workflow with reproducibility features. GEN3VA (*Gundersen et al., 2016*) is a platform for gene-expression analysis available as a web-based system.

Considering other integrative tools for transcriptome data analysis, in the literature there are different integrative approaches for analyzing transcriptomes obtained with from high-throughput technologies, such as Babelomic (*Medina et al., 2010*), RobiNA (*Lohse et al., 2012*), Expander (*Ulitsky et al., 2010*) and RMaNI (*Madhamshettiwar et al., 2013*). Most of these tools support pre-processing, filtering, clustering, functional analysis, and visualization of results. Furthermore, the tools developed are available for download or as a web interface service. Specific portals for curated bioinformatics tools can be found, for instance, on OmicTools (*Henry et al., 2014*). However, most of these tools do not support reproducibility, database management with a flexible and adequate model of representation with persistence, freedom to query the database, and function customization.

In this paper, we present GeNNet, an integrated transcriptome analysis platform that unifies scientific workflows with graph databases for determining genes relevant to evaluated biological systems. It includes GeNNet-Wf, a scientific workflow that accesses pre-loaded back-end data, pre-processes raw microarray data and conducts a series of analyses including normalization, differential expression, gene annotation, clusterization and functional annotation. During these analyses, the results are stored in different formats, e.g., figures, tables, and R workspace images. Furthermore, experiment results are stored in GeNNet-DB, which is a graph database that can be persisted. The graph database represents networks that can be explored either graphically or using a flexible query language. Finally, GeNNet-Web offers an easy-to-use web interface tool developed in Shiny for automated analysis of gene expression. The implementation follows best practices for scientific software development (*Wilson et al., 2014*). For instance, our approach uses both software containers and provenance tracking to facilitate reproducibility. This allows for reproducing, without user intervention, the computational environment (e.g., operating system, applications, libraries) and recording the applications, data sets, and parameters used in the analyses, i.e., tracking data provenance. A graph data model is used to adequately represent gene expression networks and its persistence. Also, a high-level declarative language can be used to freely query the data, existing functions can be modified and new functions added to the analytical workflow. As far as we know, GeNNet is the first platform for transcriptome data analysis that tightly couples a scientific workflow with a persistent biological (graph) database while better supporting reproducibility.

To emphasize and demonstrate the usefulness of GeNNet, we will reanalyze data from hepatocellular carcinoma (HCC) in tumor versus adjacent non-tumorous liver cells under different scenarios of use and analysis from GEO repository (Gene Express Omnibus).

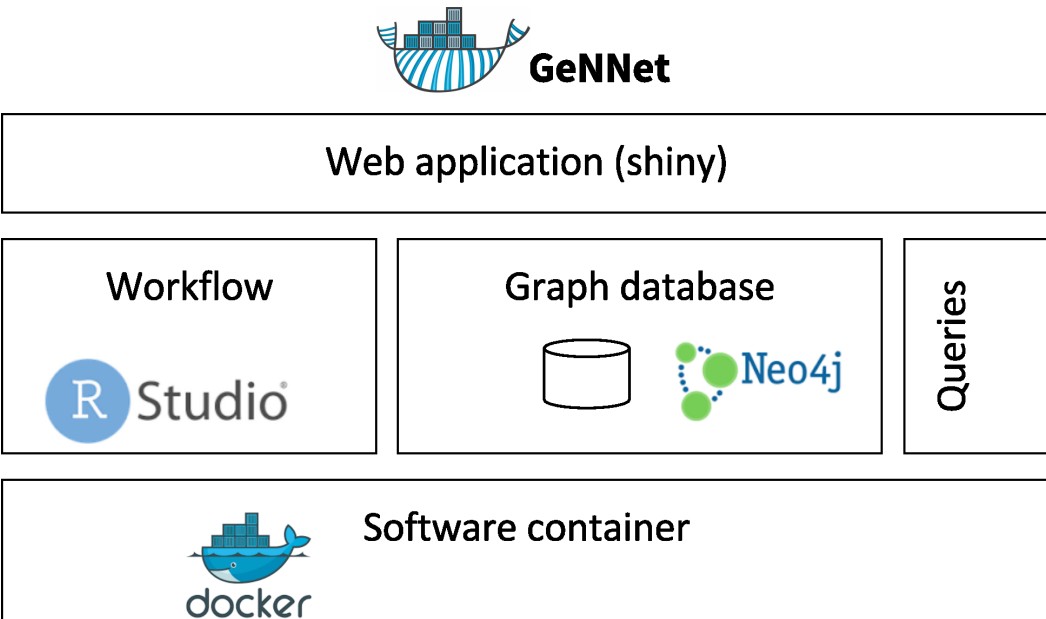

**Figure 1** GeNNet framework with its components: Scientific Workflow (GeNNet-Wf), Graph database (GeNNet-DB) and user-friendly interface (GeNNet-Web).

The first one (I) consists of executing the experiment using the user-friendly interface of GeNNet in which users choose the parameters and execute the experiment without needing to modify the lower-level scripts that compose GeNNet. The second one (II) is comprised of integrating data from different experiments that have the same hypothesis to be tested and evaluated using the RStudio IDE. The third and last one (III) uses the results of scenarios (I) and (II) to perform queries in the graph database. In (III), we highlight the use of the database persisted during the execution of GeNNet, as well as the integration of new information into it.

## MATERIALS AND METHODS

### Implementation

GeNNet innovates in its use of a graph-structured conceptual data model coupled with scientific workflow, software containers for portability and reproducibility, and a productive and user-friendly web-based front-end (Fig. 1). In the following subsections, we describe these components and functionalities in detail: scientific workflow (GeNNet-Wf), graph database (GeNNet-DB), web application (GeNNet-Web), software container, computational experiment reproducibility and experimental data.

### GeNNet-Wf workflow

GeNNet-Wf is the composition of two sub-workflows: 'Background workflow' and 'Analysis workflow' (see in Fig. 2). The data obtained by the former persists into the graph-database. The 'Analysis workflow' processes the raw dataset enriching the background data.
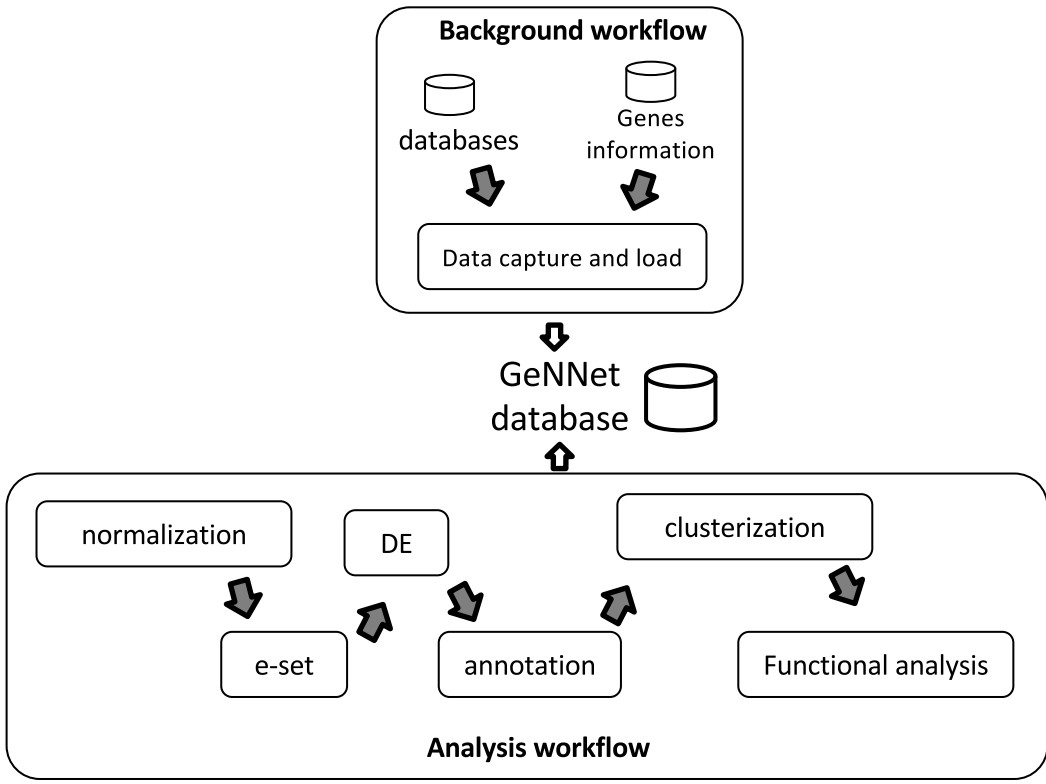

**Figure 2 Workflow scheme represented by two stages, 'Background workflow' in the top and 'Analysis workflow' in the bottom.** Results of both stages are loaded to the GeNNet database. The 'Analysis workflow' stage is shown with its different steps of the analysis. e-set is the Expression Set and DE is the Differential Expression.

### Background workflow

The GeNNet 'Background workflow' generates a database for a set of specified organisms pre-loaded into the system (Fig. 2, top). It includes genes annotated/described and their relationships, along with other associated elements, which contribute to posterior transcriptome analysis. In this version of the platform, the background data is comprised of two primary sources: (i) gene information about human, rhesus, mice and rat, obtained from NCBI annotations (*Schuler et al., 1996*); and, (ii) Protein-Protein Interaction (PPI) network, retrieved from STRING-DB (*Franceschini et al., 2013*) (version 10). All genes imported from NCBI become nodes in the graph database and some of the primary information associated with them (such as symbol, entrezId, description, etc.) are modeled as node properties. The information derived from STRING-DB PPI become edges ('neighborhood', 'gene fusion', 'co-occurrence', 'co-expression', 'experiments', 'databases', 'text-mining' and 'combined score'). This layer of data is added to the graph database during the construction of the GeNNet container (*Software container* subsection). More detail about the representation and implementation can be found in section *GeNNet-DB graph database*.

### Analysis workflow

The 'Analysis workflow' stage is comprised of the execution of a series of tools and libraries to analyze the transcriptome data uploaded by the user in conjunction with the data generated by the 'Background workflow' (*Fig. 2*, bottom). This module was written in R using several packages mainly from the Bioconductor (*Dudoit, Gentleman & Quackenbush, 2003*) and CRAN repositories. The steps are detailed next.

*Normalization.* This step consists in normalizing the raw data from an informed Affymetrix platform using either RMA (*Irizarry et al., 2003*) or MAS5 methods, both available in the affy (*Gautier et al., 2004*) package. During this step, some quality indicator plots are generated (as boxplot of probe level, Spearman correlation, and density estimates) as well as a normalized matrix (log-normalized expression values).

*e-set.* In this step, data about the experimental design should be added along with log-normalized expression values. This generates an ExpressionSet (eSet) object, a data structure object of the S4 class used as a base in many packages developed in Bioconductor transcriptome analysis (*Falcon, Morgan & Gentleman, 2007*) . This format gives flexibility and access to existing functionality. The input file must be structured using mainly two columns: a column named SETS for the experimental design, and a column called SAMPLE_NAME for the names of the files containing raw sample expression matrix data.

*Filtering/Differential expression inference.* Differential expression (DE) inference analysis allows for the recognition of groups of genes modulated (up- or down-regulated) in a biological system when compared against one or more experimental conditions. In many situations, this is a core step of the analysis, and there is a great diversity of experimental designs (such as control versus treatment, consecutive time points, etc.) allowing the inference. In our platform, we use the limma package to select the DE genes (*Smyth, 2004*) on single-factor experimental designs based on a gene-based hypothesis testing statistic followed by a correction of multiple testing given by the False Discovery Rate (FDR) (*Kendall & Bradford Hill, 1953*). Furthermore, a subset of DE genes can be selected based on up- and down-regulation expressed as an absolute logarithmic (base 2) fold-change (logFC) threshold. The latter can be set-up by the user, as described in *Scenario I—Experiment user-friendly interface*. Results of this step are displayed as Volcano plots and matrices containing the DE genes.

*Annotation.* The annotation step consists of annotating the probes for the corresponding genes according to the Affymetrix platform used in the experiment.

*Clusterization.* This step consists in analyzing which aggregated genes have a similar pattern (or level) of expression. We incorporated clusterization analysis including hierarchical methods, $k$-medoids from the package PAM (Partitioning Around Medoids) (*Reynolds et al., 2006*) and WGCNA (Weighted Gene Coexpression Network Analysis) (*Langfelder & Horvath, 2008*).

*Functional analysis.* Genes grouped by similar patterns enable the identification of over-represented (enriched) biological processes (BP). In our approach, we conducted enrichment analyses applying hypergeometric tests (with $p$-value $< 0.001$) as implemented in the GOStats package (*Falcon & Gentleman, 2007*). Ontology information for the gene is extracted from the Gene Ontology Consortium database (*Ashburner et al., 2000*). The universe is defined as the set of all genes represented on a particular Affymetrix platform, or, in the case of multiple platforms in a single experiment design, the universe is defined as the common and unique genes in among all Affymetrix platforms. The subset is defined either by the set of diferentially expressed (DE) genes between a test and a control condition (control versus treatment design) or by the union of the DE genes selected among the pairwise comparisons among groups in all other single-factor experimental designs. Although functional analyses can lead to biased results, as presented in *Timmons, Szkop & Gallagher (2015)*, we have added more restrictive cut-off with the purpose of reducing the detection bias of our platform.

### Execution

GeNNet is designed to automatically execute the workflow through the web application interface (accessed via http://localhost:3838/gennet, when the software container is running). However, users that intend to implement new functions or even execute the workflow partially, can use the RStudio server interface in GeNNet (accessed via http://localhost:8787 after starting the software container). More details are available in Supplemental Information.

## GeNNet-DB graph database

Although a NoSQL database has no fixed schema, we defined an initial graph model to help and guide the GeNNet-DB (Fig. 3). GeNNet database (GeNNet-DB) structure is defined on the Neo4j database management system, a free, friendly-to-use and with broad community support graph database, with its nodes, edges, and relationships. Vertices and edges were grouped into classes, according to the nature of the objects. We defined the labels as GENE, BP (Biological Process), CLUSTER, EXPERIMENT, ORGANISM, and a series of edges as illustrated in Fig. 3. In the GeNNet platform there is an initial database defined by interactions between genes as described in *Background preparation* section. During the execution of GeNNet-Wf, using Shiny or RStudio, new nodes and connections are formed and added to the database. The resulting information is stored in the graph database using the RNeo4j package available at: (https://cran.r-project.org/web/packages/RNeo4j). It can also be accessed directly through the Neo4j interface (accessed via http://localhost:7474). It is possible to query and access the database in this interface using the Cypher language, a declarative query language for Neo4j, or Gremlin, a general-purpose query language for graph databases. These query languages allow for manipulating data by updating or deleting nodes, edges, and properties in the graph. Querying also allows for exploring new hypotheses and integrating new information from different resources that are related to the targeted experiment. GeNNet-DB is persistent, and the resulting database is exported to a mounted directory. Its contents can be loaded to a similar Neo4j installation. For further details, one can read the Neo4j manual.

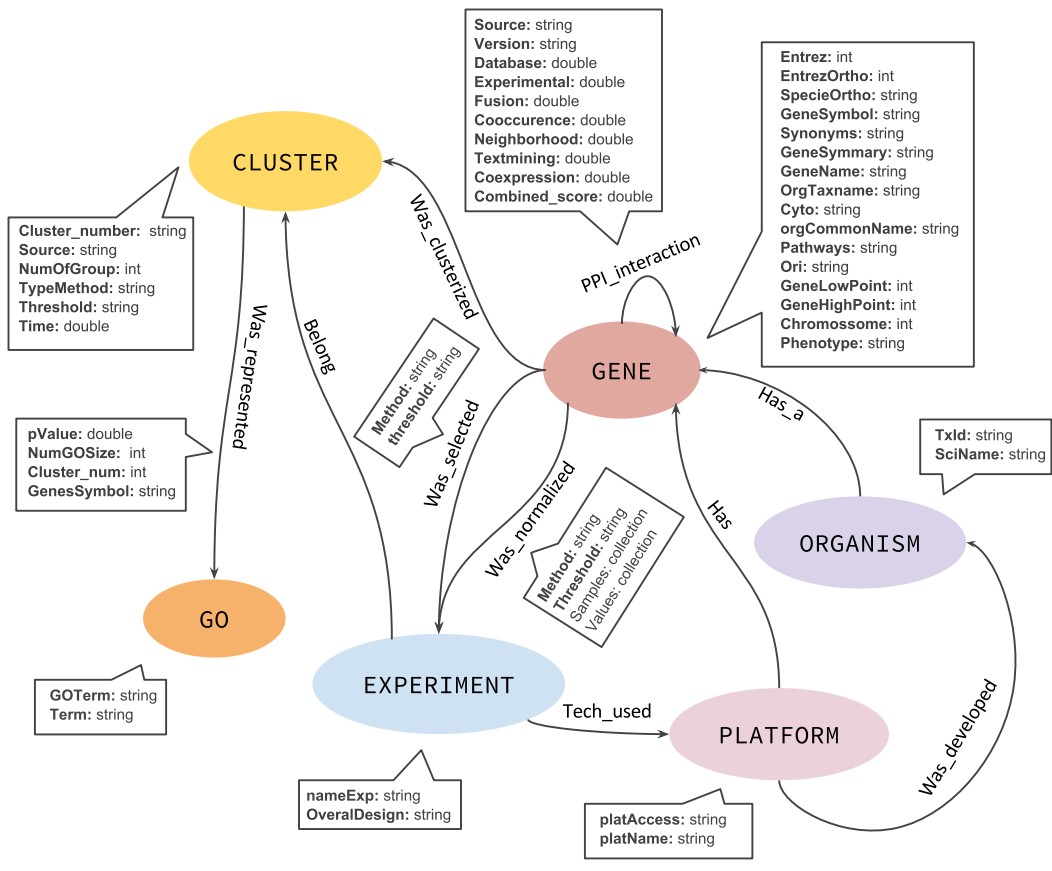

**Figure 3** **The graph model representing the nodes (in oval), relationships (arrows).** The descriptive boxes showing the mainly properties in nodes and edges.

## GeNNet-Web application

GeNNet-Web provides a user-friendly way to execute GeNNet-Wf. We developed an easy-to-use layout for providing the parameters and automatically executing all steps of the workflow experiment. The application was implemented using the Shiny library for R. This library allows for turning an R script that implements some analysis into a web application in a convenient manner. Shiny has a library of user interface elements can be used for entering input data and parameters, and for displaying the output of an R script. The parameters comprise the input of the web application, which includes: descriptors for experiment name and overall design; type of normalization; differential expression settings; experiment platform and organism; and clusterization method. After executing GeNNet-Wf, GeNNet-Web allows for easy retrieval and visualization of its outputs, which are given by a heatmap, graph database metrics (e.g., the number of nodes, the number of edges and relationships between nodes), and the list of differentially expressed genes selected. In addition to the outputs generated in the web application, the underlying workflow creates the output files described in subsection *GeNNet-Wf workflow*.

## Software container

A Docker software container was built containing GeNNet and all its required libraries and dependencies. This enables users to download a single software container that includes all the components of GeNNet and instantiate this environment independently in any host that runs an operating system supported by Docker. The software container was successfully tested on CentOS Linux 7, Ubuntu Linux 14.04, MacOS X 10.11.6 and Windows 10 hosts. The software container for GeNNet, specified in a script named 'Dockerfile', was built according to the following steps: (i) The operating system environment is based on Debian GNU/Linux 8 with software packages required by GeNNet, such as R (v. 3.3.1), installed from the official Debian repositories; (ii) The R software and the packages required by GeNNet, installed from the CRAN repository; (iii) RStudio (v. 1.0.44) server and the Neo4j (Community Edition v.3.0.6) graph database, installed from their respective official repositories; (iv) Supporting data sets, such as PPI, loaded to the graph database; (v) GeNNet-Wf, implemented in R, installed in RStudio; (vi) Shiny, a web application server for R, installed from its official repository. GeNNet-Web, which calls GeNNet-Wf, is loaded to Shiny.

## Computational experiment reproducibility

Reproducibility is accounted in GeNNet in two aspects. Firstly, the platform provides a provenance trace record generated by the RDataTracker package (*Lerner & Boose, 2015*) for R. The trace contains the activities executed by the workflow and the data sets consumed and produced by them. This trace is exported to a persistent directory. Secondly, the adoption of software containers allows for using the same environment (operating system environment, libraries, and packages) every time GeNNet is instantiated and used. Both the provenance trace and the preservation of the execution environment with software containers significantly help the computational experiment reproducibility since users can retrieve from the former the parameters and data sets used in analyses and, from the latter, re-execute them in the same environment, as provided by the GeNNet software container.

## Experimental data—use case scenarios

To illustrate the flexibility of GeNNet, we will conduct an experiment of re-analysis of HCC, considered the most common type of liver cancer. The HCC is highly complex, and the main risk factors are associated with prolonged abusive alcohol consumption and persistent infection of HBV (Hepatitis B Virus) and HCV (Hepatitis C Virus) (*Siegel, Miller & Jemal, 2017*). We performed the re-analysis of microarray experiments deposited in the GEO repository and, to facilitate the understanding of GeNNet, we separated this case study in three different scenarios. The first one is to analyze the data using the friendly web interface for GeNNet developed in Shiny (described in '*GeNNet-Web application*'). The second scenario is to integrate an additional independent experiment to the data using the RStudio interface to create and modify their functions. The last scenario is to perform queries in the graph database generated during the execution and analysis of the experiment, highlighting the range of possibilities of the system we developed.

## RESULTS AND DISCUSSION

### Scenario I—experiment user-friendly interface

As an example of a specific and more detailed case study, we re-analyzed a gene expression experiment from HCC obtained from the transcriptome repository GEO (*Barrett et al., 2013*) with accession number GSE62232 (*Schulze et al., 2015*). The study used the Affymetrix Human Genome U133 Plus (GPL570) and contained 91 samples, of which 81 samples are from HCC tumors and 10 from adjacent non-tumorous liver tissues.

Data was normalized using the MAS5 method and the differentially expressed gene selection criteria were FDR <0.05 and absolute log2(Fold-Change) >1. The initial threshold values chosen are the most used and recommended in the literature but the threshold values can be adjusted. The genes were clustered using the Pearson correlation method as a measure of dissimilarity. Next, the clusters were associated with biological functions through the hypergeometric test (with *p*-value < 0.001 as threshold). All parameters were configured using the friendly interface built in Shiny as shown in Fig. 4 and accessed via http://localhost:3838/gennet. As a result, 3,356 differentially expressed genes were obtained, and 661 ontological terms were represented (*p*-value < 0.001). A major part of the information arising from the analytical process was incorporated to GeNNet-DB. Besides the database, the results were exported to different formats such as figures (heatmaps, boxplots, etc.), tables and provenance (Fig. 5).

### Scenario II—RStudio environment in meta-analysis

In this scenario we explored the flexibility introduced by the integration of RStudio in our platform. Its availability enables more experienced users to extend existing functionality with new analyses over available data. In this scenario, we explore one example of such flexibility with a meta-analysis approach in which we combine results from different experiments. Meta-analysis experiments combine microarray data from independent yet similarly designed studies allowing one to overcome their variations, and ultimately increasing the power and reproducibility of the transcriptome (*Ewald et al., 2015*) analysis. We added a study with experimental design performed on the data described in the previous section. We used HCC data containing 18 tumor samples versus 18 adjacent non-tumorous liver tissues from *Wang et al. (2014)*. The experiment was carried out with the Affymetrix Human Genome U133 platform and deposited in GEO under accession number GSE60502. The Fig. 6 shows the access via RStudio (accessed via: http:localhost:8787).

This scenario of use requires more advanced users in the R language. We exemplify the addition of an experiment to enhance the flexibility of our platform by making the analysis more robust and integrative between complex experiments as in cancer studies. However, the user can modify or even add a function by generating new analyses from GeNNet.

### Scenario III—querying and adding relationships

Biological information is typically highly connected, semi-structured and unpredictable. The results obtained from the GeNNet analysis are stored in a graph database during the execution of the workflow. The database can be accessed via http://localhost:7474 using the Cypher declarative query language with direct access to the database, we formulated

## GeNNet Platform

**Upload Phenodata in CSV File [OPTIONAL]**

| Browse... | No file selected |
|---|---|

**Separator**

○ Tab
● Comma

Information about the experiment

**Experiment name**

GSE62232

For example, the name of GEO accession number.

**Overall design**

HCC liver tumors corresponding to 81 patients. In all cases

Normalization Parameters

**Type of Normalization**

MAS5

Set Parameters to apply Differential Expression

**Log2(Fold-Change) in abs:**

| 0.5 | 1 | | | | | | | | | | 10 |

0.5   1.5   2.5   3.5   4.5   5.5   6.5   7.5   8.5   9.5   10

**FDR:**

| 0.001 | | | | 0.05 | | | | | | 0.1 |

0.001   0.011   0.021   0.031   0.041   0.051   0.061   0.071   0.081   0.091   0.1

Platform Parameters

**Chose Platform**

GPL570: [HG-U133_Plus_2] Affymetrix Human Genome U133 Plus 2.0 Array

Welcome to GeNNet!     CEL Archives Data     PhenoData matrix     Heatmap     Interactive Heatmap

Functional Analysis     Topologies     Database scheme     Graph-DB metrics     Cite us

Show 25 entries                                                    Search:

| SAMPLE_NAME | Type | SETS |
|---|---|---|
| GSM1523329_JZN_0469N_U133_2.CEL.gz | CHC469N | adjacent |
| GSM1523330_JZN_0562N_U133_2.CEL.gz | CHC562N | adjacent |
| GSM1523331_JZN_0591N_U133_2.CEL.gz | CHC591N | adjacent |
| GSM1523332_JZN_0591N_U133_2.2.CEL.gz | CHC591N.rep2 | adjacent |
| GSM1523333_JZN_0591N_U133_2.3.CEL.gz | CHC591N.rep3 | adjacent |
| GSM1523334_JZN_0932N_U133_2.2.CEL.gz | CHC932N | adjacent |
| GSM1523335_JZN_0932N_U133_2.CEL.gz | CHC932N.rep2 | adjacent |
| GSM1523336_JZN_0934N_U133_2.CEL.gz | CHC934N | adjacent |
| GSM1523337_JZN_0934N_U133_2.2.CEL.gz | CHC934N.rep2 | adjacent |
| GSM1523338_JZN_0934N_U133_2.3.CEL.gz | CHC934N.rep3 | adjacent |
| GSM1523339_JZN_0012T_U133_2.CEL.gz | CHC012T | Tumor |
| GSM1523340_JZN_0228T_U133_2.CEL.gz | CHC228T | Tumor |
| GSM1523341_JZN_0235T_U133_2.CEL.gz | CHC235T | Tumor |
| GSM1523342_JZN_0250T_U133_2.CEL.gz | CHC250T | Tumor |
| GSM1523343_JZN_0327T_U133_2.CEL.gz | CHC327T | Tumor |
| GSM1523344_JZN_0434T_U133_2.CEL.gz | CHC434T | Tumor |
| GSM1523345_JZN_0891T_U133_2.CEL.gz | CHC891T | Tumor |
| GSM1523346_JZN_1010T_U133_2.3.CEL.gz | CHC1010T | Tumor |
| GSM1523347_JZN_1044T_U133_2.CEL.gz | CHC1044T | Tumor |
| GSM1523348_JZN_1185T_U133_2.CEL.gz | CHC1185T | Tumor |
| GSM1523349_JZN_0050T_U133_2.CEL.gz | CHC050T | Tumor |
| GSM1523350_JZN_0080T_U133_2.CEL.gz | CHC080T | Tumor |
| GSM1523351_JZN_0203T_U133_2.CEL.gz | CHC203T | Tumor |
| GSM1523352_JZN_0239T_U133_2.CEL.gz | CHC239T | Tumor |
| GSM1523353_JZN_0339T_U133_2.CEL.gz | CHC339T | Tumor |
| SAMPLE_NAME | Type | SETS |

Showing 1 to 25 of 91 entries                    Previous   1   2   3   4   Next

**Figure 4    User-friendly interface in GeNNet.** Left-hand side showing parameters settings and right-hand side showing some tabs highlight Pheno-Data matrix uploaded.

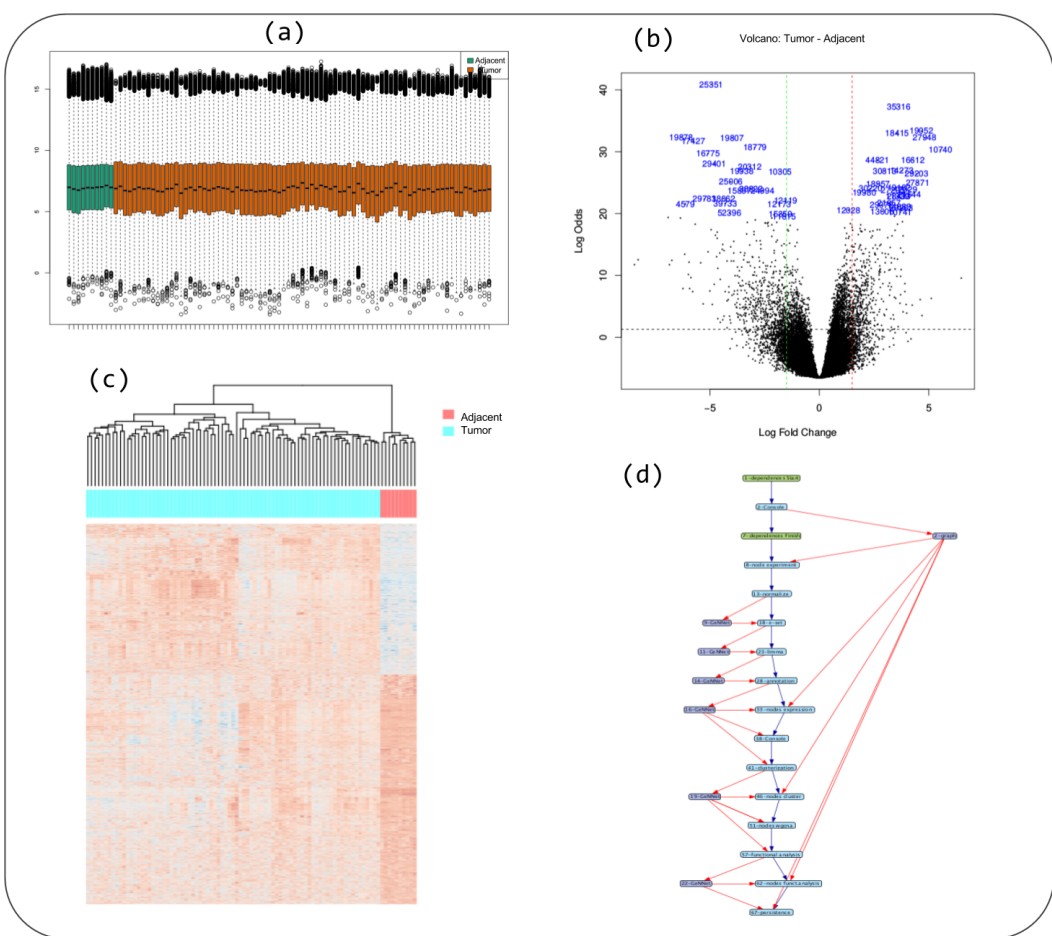

**Figure 5** Some captions of figures generated during workflow execution: (A) boxplot of quality indicator, (B) volcano plot, (C) heatmap of genes differentially expressed in tumor versus adjacent, and (D) the provenance trace of a GeNNet-Wf execution is represented as a data derivation graph (DDG).

some demonstration queries using as an example the dataset analyzed above. The database generated during GeNNet-Wf execution facilitates data representation as interaction networks, in an approach that allows for exploring a great variety of relationships among its composing entities, besides making new insights for subnetwork exploration possible. Depending on the type of these interactions, different kinds of networks and topologies can be defined and analyzed. Through the data representation used in GeNNet-DB, traversal queries are possible. We illustrate typical examples in which the user just needs to query GeNNet-DB to solve them.

*Query 1*: What are the existing relationships among nodes in the database?

This is a simple query that returns all existing relationships among different node labels and types. The result of the query was represented as a graph in Fig. 7 retrieved the graph model as exemplified in Fig. 3.

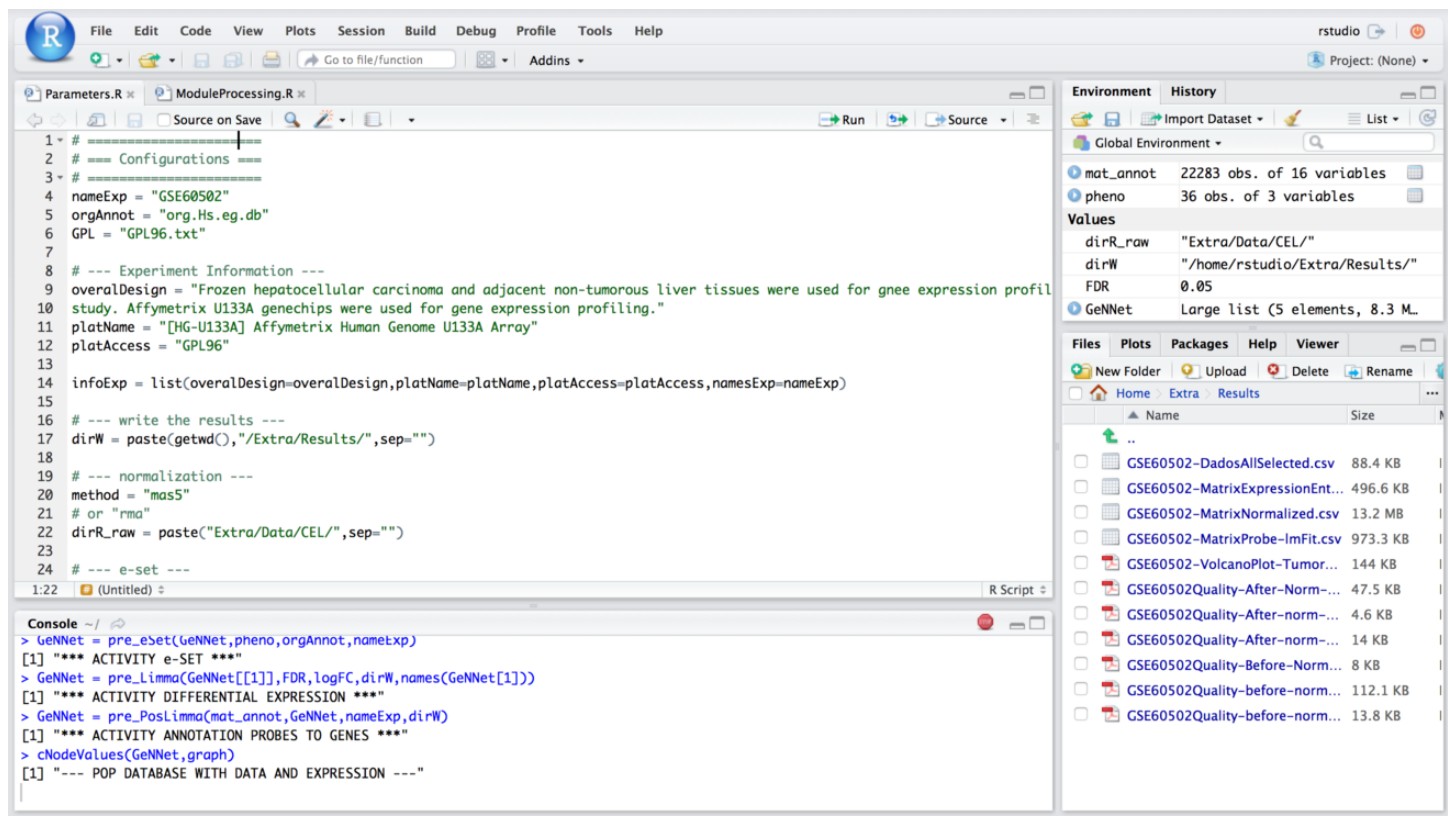

**Figure 6** Example of access mode to RStudio where it is possible to modify the predefined functions.

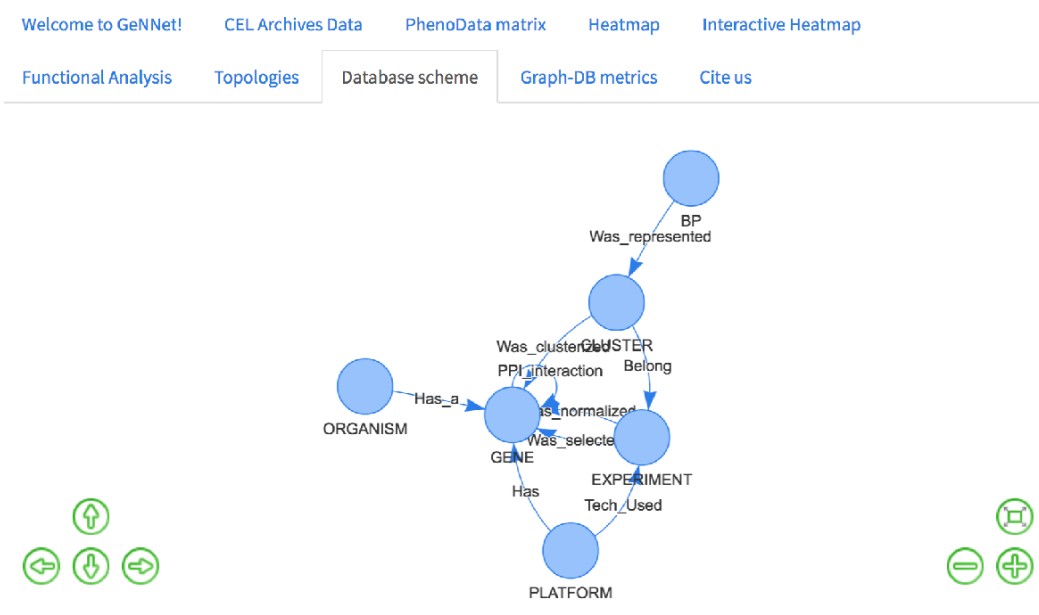

**Figure 7** Database schema with all the existing nodes and relationships.

**Table 1  Result showing the top 10 gene DE by PPI in experiment GSE62232.** These genes are known as hubs and may be associated with important pathways in the experimental context analyzed.

| Genes selected | cgn[a] | BP associated |
|---|---|---|
| CDK1 | 89 | |
| SRC | 92 | |
| PLK1 | 83 | regulation of tau-protein kinase activity; L-cysteine metabolic process; negative regulation of natural killer cell differen-tiation; response to lipopolysaccharide; positive regulation of angiogenesis; regu-lation of lipid metabolic process |
| JUN | 73 | |
| BIRCS | 68 | |
| AURKB | 68 | |
| FOS | 66 | |
| PCNA | 61 | |
| ADCYS | 60 | |
| POMC | 60 | |

**Notes.**
[a] number of connected genes.

```
MATCH (a)−[r]−(b)
WHERE labels(a) <>[] AND labels(b) <>[]
RETURN DISTINCT head(labels(a)) AS This,
               type(r) as Relation,
               head(labels(b)) as To
```

*Query 2*: Which nodes of type GENE were DE and present the highest number of connections associated to the protein interaction networks (PPI) according to a combined score threshold of >0.80? Among these selected nodes, what are the clusters and associated biological processes?

Some common and important topological metrics in biological networks include: degree, distance, centrality, clustering coefficient. In this work, we use the degree metric $k_i$ of a node $n_i$, defined as the number of edges that are adjacent ($a_{ij}$) to this node, which is given by:

$$k_i = \sum_{j \in V} a_{ij}. \tag{1}$$

We use the Cypher query language to find the most connected DE genes in the network that establish known connections to the PPI network, having a high attribute value for the combined interaction score (provided by PPI association of protection interaction database STRING-DB). For these genes we computed the co-expression cluster and, subsequently, the biological processes attributed to these clusters. One can observe that the query is expressed concisely for answering a relatively complex topological question. The resulting DE genes are displayed in Table 1.
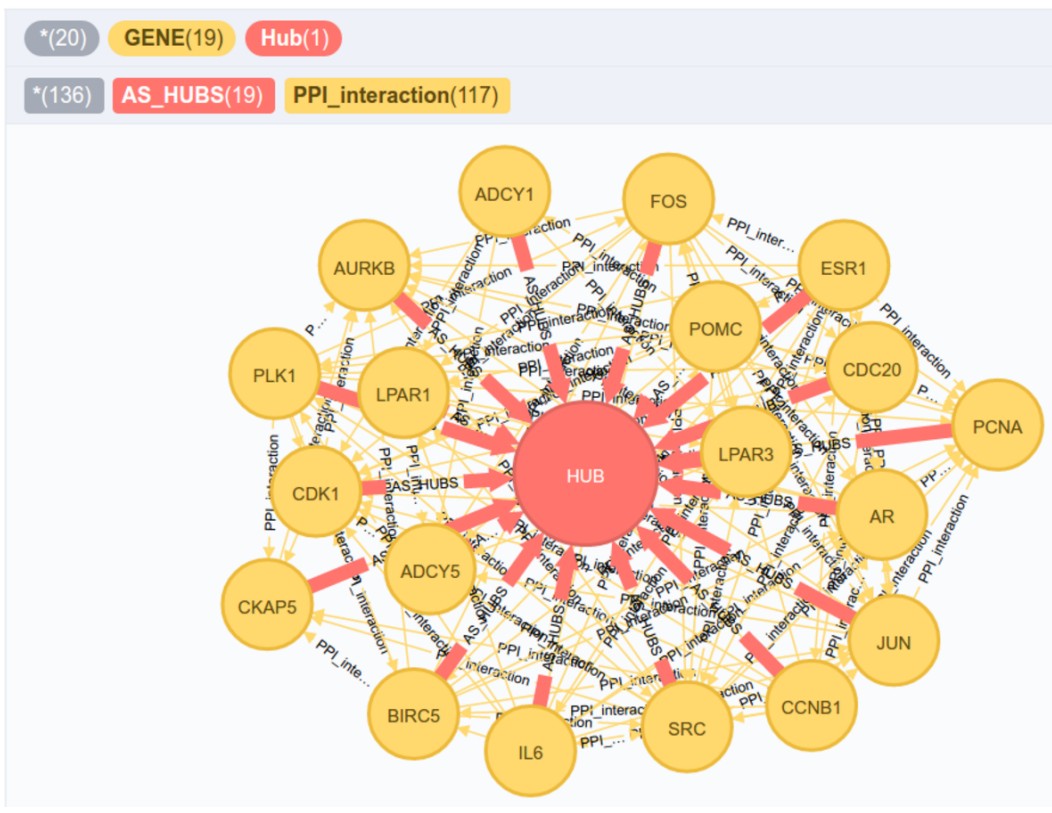

**Figure 8  New nodes and edges added to the graph database.** The genes that were highly connected according to query 3 were directed to a node of type HUB.

```
MATCH (e:EXPERIMENT)−[s:Was_selected]−>
      (g:GENE)−[p:PPI_interaction]−(h:GENE)−
      [:Was_clusterized]−(c:CLUSTER)−
      [:Was_represented]−(b:BP)
WHERE p.combined_score > 0.80
RETURN distinct g.symbol,
      COLLECT(distinct(h.symbol)) AS genes,
      COLLECT(distinct(b.Term)) AS BP,
      COUNT(distinct h) AS score
      ORDER BY score DESC LIMIT 10
```

One of the main advantages of using the data model adopted in GeNNet is the availability of data and information that can be easily done without changing the data model. New nodes may add information such as metadata of samples (e.g., information on a patient's eating habits) or new edges may add new relationships (e.g., genes co-expressed in different methods used) or even both (e.g., addition of a database on microRNA interactions connected to existing genes in the database). In the example below, we add a HUB-like node from the result obtained in query 2. Through the CREATE clause, after obtaining
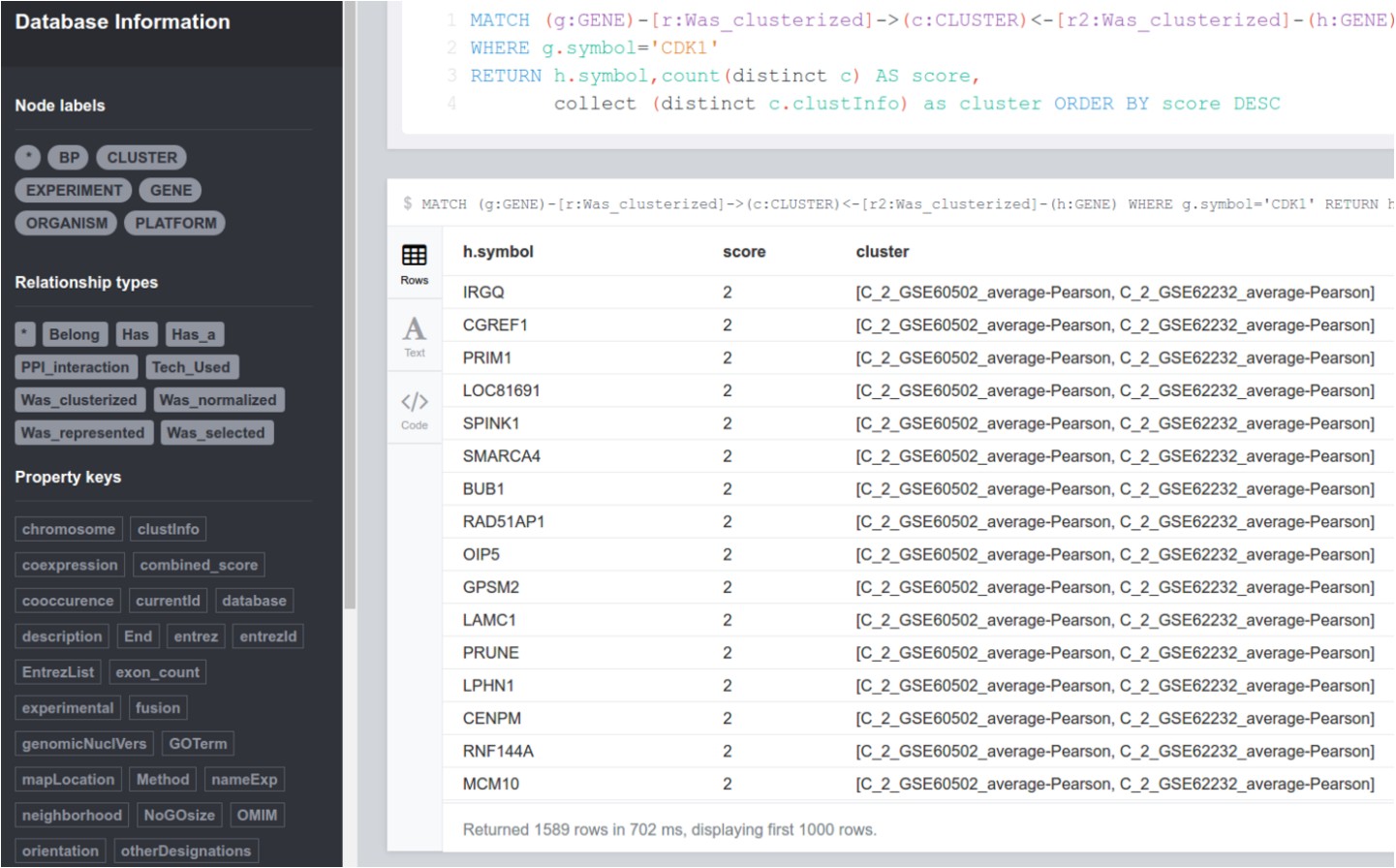

**Figure 9** Result of query 4 in the Neo4j access interface displaying the genes co-expressed with CDK1 in the different experiments deposited in GeNNet-DB.

the selected genes, a new node and edges were created (Fig. 8). These queries demonstrate the flexibility of the database in adding new information that can be generated through existing data in GeNNet-DB.

*Query 3*: New node and edges inserted from the result of the previous query.

```
MATCH (e:EXPERIMENT)−[s:Was_selected]−>
        (g:GENE)−[p:PPI_interaction]−(h:GENE)−
        [:Was_clusterized]−(c:CLUSTER)−
        [:Was_represented]−(b:BP)
WHERE p.combined_score > 0.80
WITH DISTINCT g, COUNT(distinct h) AS score
WHERE score > 50 WITH collect(g) AS gs
CREATE (hub:Hub {name: 'HUB'})
WITH gs, hub UNWIND gs AS g
CREATE (g)−[:AS_HUBS]−>(hub)
RETURN *
```

*Query 4*: Given different experiments, which genes are co-expressed with a differentially expressed gene, for instance, gene 'CDK1'?

Through this query, we can know which genes are co-expressed with CDK1 ranked in descending order on the number of experiments analyzed and deposited in the database. As a result of this query, we obtain that 326 genes appear co-expressed with gene CDK1 in both experiments analyzed in scenarios I and II (Fig. 9).

```
MATCH (g:GENE)−[r:Was_clusterized]−>(c:CLUSTER)<−
[r2:Was_clusterized]−(h:GENE)
WHERE g.symbol='CDK1'
RETURN h.symbol,count(distinct c) AS score,
     collect (distinct c.clustInfo) as cluster
ORDER BY score DESC
```

## CONCLUSION, UPDATES AND FUTURE WORK

The platform presented in this work is the first one to integrate the analytical process of transcriptome data (currently only available for microarray essays) with graph databases. The results allow for testing previous hypothesis about the experiment as well as exploring new ones through the interactive graph database environment. It enables the analysis of different data coming from Affymetrix platforms on humans, rhesus, mice and rat. GeNNet will be periodically updated, and we intend to extend the modules to include analyses of RNA-Seq and miRNA. We will incorporate additional experimental designs for DE and improve the execution time of the analyses. Moreover, we intend to add other model organisms to the background data, such as *Arabidopsis thaliana* and *Drosophila melanogaster*.

GeNNet-Web offers an interface that accommodates both experienced and inexperienced users. For the latter, the interface provides various filtering and parameter setup opportunities, in addition to some pre-defined queries. For more advanced users a plain query interface is provided so that more tailored analysis can be expressed. Due to the free access to GeNNet, we rely on the feedback of the community for improving the tool. The distribution of the platform in a software container allows not only for executing it on a local machine but also for easily deploying it on a server and making it available on the Web.

### Funding

This work has been supported by CAPES (Coordenação de Aperfeiçoamento de Pessoal de Nível Superior) and CNPq (Conselho Nacional de Desenvolvimento Cientíifico e Tecnológico) funding. The funders had no role in study design, data collection and analysis, decision to publish, or preparation of the manuscript.

## Grant Disclosures

The following grant information was disclosed by the authors:
CAPES (Coordenação de Aperfeiçoamento de Pessoal de Nível Superior).
CNPq (Conselho Nacional de Desenvolvimento Cientíifico e Tecnológico).

## Competing Interests

The authors declare there are no competing interests.

## Author Contributions

- Raquel L. Costa conceived and designed the experiments, analyzed the data, contributed materials/analysis tools, wrote the paper, prepared figures and/or tables, reviewed drafts of the paper.
- Luiz Gadelha conceived and designed the experiments, wrote the paper, reviewed drafts of the paper.
- Marcelo Ribeiro-Alves contributed materials/analysis tools, wrote the paper, reviewed drafts of the paper.
- Fábio Porto wrote the paper, reviewed drafts of the paper.

## Data Availability

   Github: https://github.com/raquele/GeNNet.
   Docker hub: https://hub.docker.com/r/quelopes/gennet/.

## Supplemental Information

Supplemental information for this article can be found online at http://dx.doi.org/10.7717/peerj.3509#supplemental-information.

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
