# Peer review of "GeNNet: an integrated platform for unifying scientific workflows and graph databases for transcriptome data analysis"

_PeerJ, doi:10.7717/peerj.3509_

## Round 0.1 · original submission · Major Revisions

The paper you submitted is clear and quite interesting, however there are some major points to be solved, as highlighted by the attached referees' comments. The manuscript requires a critical review with some additional information: in order to be published the data should be robust, scientifically sound, and verified.

·

Basic reporting

The paper is overall clear and flows well. The introduction is smooth and addresses well the methodology and the results described by the authors. However, the English can be improved in some places, as well as some scientific definitions and explanations and figures annotations/legends.
I am listing here some of the areas of improvement:
1- GeNNet evidently analyzes microarray data and not all types of transcriptomic data. For that, I suggest to specify this in the title.
2- Lines 54-60: the authors focus on highly-weighed nodes and support in gene network and ignores low-weighed ones can you explain why? The same for positive and highly significant correlations, what about negative and highly significant correlations among group of genes?
3- line 88 – 89: I suggest to add a more recent citation regarding data integration, these are common nowadays (see Conesa et al. Genome Biology; DOI 10.1186/s13059-016-0881-8)
4- line 105: what is the definition of ‘high-level query language’? This is not a common concept for non-IT expert users.
5- line 133: it is not clear what GeNNet-Wf really does. Are the pre-loaded data the persistent database or the pre-processed raw microarray data become the persistent database?
6- line 135: what do you mean by annotation alone, given that you mention ‘functional annotation’ later on.
7- line 146: where is the data comes from, it is mentioned only in the section ‘Results and Discussion’, it is worthy to mention in the first occasion.

Experimental design

GeNNet is an assembly of already existing tools especially R scripts/packages able to analyze microarray data. Building of GeNNet-DB is stated by the author as an important object, which can be re-used for additional and different experimental data. Implementation steps are described enough but some details are missing. For instance, the Filtering/Differential expression inference (line 194 ), It is not clear if FDR and fold change threshold are dynamic and can be changed by the user. Another important point regards the back-end data which are available only for 4 organisms, is it possible through Rstudio to load background data for additional or other organisms such as plants?
Note also that supplementary materials needs some revisions, especially regarding parameter settings. It is not evident which parameters and thresholds are pre-defined by the system and those that can be user-defined and why. Moreover, from developer side, is it possible to change models within R scripts or upload new R package in the system. This gives an idea about the flexibility of the system defined as workflow management.
Another important point are the figures, they should be better annotated, for instance:
- Figure 1 legend would be clearer if the meaning of the abbreviations are stated again not only in the text, for instance scientific workflow (GeNNet-Wf)
- Figure 2 legend: please add the abbreviations, e.g e-set?
- Figure 4 legend to be changed: on left hand side there are no parameters settings boxes, and on the right hand there are only tabs of obtained results.
- Figure 5: not clear very blurred.

Validity of the findings

The idea behind the present study is promising and the used data are interesting as a test case, but in my opinion the data lack of experimental (in-silico or in-vitro) validations. Raw data from GEO were used to run the workflow but we do not know whether the obtained results are correct. This is an important point especially in scenario III, in which the authors query the database to get biological conclusions. I recommend to add precision and accuracy analysis based on some Gold Standard micoarray data or some positive and negative control data set.

Additional comments

The study is interesting and covers an important area for transcriptomic analyses. I would recommend to read carefully the manuscript again and add additional information and some analyses to improve its scientific strength. In my opinion GeNNet is not easy to use by non-experts especially the query building part (described in scenario III). So probably, it is enough to state that GeNNet is made for advanced bioinformatics users and/or computational biologist. In addition, adding tools targeting other transcriptomic analyses (i.e. RNAseq, miRNA) would need a substantial flexibility to dynamically assemble new workflows within GeNNet, so if that is possible it is convenient to mention that clearly.

Reviewer 2 ·

Basic reporting

- Regarding the language, the paper was clear to understand.

- I missed a number of references regarding workflow management systems, like Galaxy (Giardine et al 2005) and Tavaxy (Abouelhoda et al 2012).

- I missed also a number of important references to systems (desktop and web-based) for microarray data analysis and gene set enrichment analysis.

Experimental design

- The experimental part was a kind of demonstration to the use of the tool and did measure any advantage in comparison to other software tools in terms of accuracy of results or efficiency of processing.

Validity of the findings

The lack of comparison to other systems is a limiting factor.

Additional comments

The paper introduces a system for transcriptome data analysis and representation of the results using graph databases to evaluate efficient query of the results.
I have the following comments:

- The implemented software is a combination of script-based pipeline for analysis of microrray datasets, using well known open source tools and packages. The results of the analysis are either visualized using R-shiny-based interface or inserted into a graph database for further query.
The whole system is included in a docker container.
These steps are practised by any bioinformatician these days; where the composition of analytical pipelines and use of databases to manage the results within a docker container became straightforward tasks. I could not identify a technical challenge the authors could address.
The authors use usual tools and packages with straightforward use of database enginers.

- The paper lacks good survey of available tools and their contribution in this regard. For example, many tools establish a network view for gene-gene relationships; usually after the enrichment step.
Is there any difference of what is already available..

- The authors suggest the use of NoSQL graph based database systems, but did not justify the need for this. After running enrichment analysis, usually a mnageable number of genes remains in the list to be visualized. So more discussion to justify the use of advanced tool in terms of big data is required.

- I have some concerns regarding the use of two words in the paper which a little bit misleading:
a) transcriptime; which may give the impression that the tool in the paper can handle NGS data starting from raw sequence reads, which seems not be the case. I see that the input to the tool is usually some expression values (mostly from microarray results).

b) workflow systems: the authors do not provide a new workflow management system. They have a pipeline composed of a sequence of tools running after another, and do not have a generic workflow engine.

- I wonder why the authors did not use an existing workflow enginer to implement the pipeline; then identify some gaps to justify the development of their tools. This is what usually application scientist would do.

---

## Round 0.2 · accepted · Accept

The paper improved a lot after the revision.

·

Basic reporting

The manuscript has improved much, all questions were answered and ambiguities are more clear. It is also promising that the authors will maintain and add additional updated data to their GeNNet platform.
Just quickly review the English in the main manuscript.

Experimental design

It is much more clear now.

Validity of the findings

no comment

Additional comments

Please correct the legend of the figures in supplementary materials, you still have the figure called in Portuguese (figura..).